# Recent Advances of Organ-on-a-Chip in Cancer Modeling Research

**DOI:** 10.3390/bios12111045

**Published:** 2022-11-18

**Authors:** Xingxing Liu, Qiuping Su, Xiaoyu Zhang, Wenjian Yang, Junhua Ning, Kangle Jia, Jinlan Xin, Huanling Li, Longfei Yu, Yuheng Liao, Diming Zhang

**Affiliations:** 1Guangdong Provincial Key Laboratory of Industrial Surfactant, Institute of Chemical Engineering, Guangdong Academy of Sciences, Guangzhou 510075, China; 2Research Center for Intelligent Sensing Systems, Zhejiang Laboratory, Hangzhou 311100, China

**Keywords:** organ-on-a-chip, tumor microenvironment, microfluidics, cancer modeling

## Abstract

Although many studies have focused on oncology and therapeutics in cancer, cancer remains one of the leading causes of death worldwide. Due to the unclear molecular mechanism and complex in vivo microenvironment of tumors, it is challenging to reveal the nature of cancer and develop effective therapeutics. Therefore, the development of new methods to explore the role of heterogeneous TME in individual patients’ cancer drug response is urgently needed and critical for the effective therapeutic management of cancer. The organ-on-chip (OoC) platform, which integrates the technology of 3D cell culture, tissue engineering, and microfluidics, is emerging as a new method to simulate the critical structures of the in vivo tumor microenvironment and functional characteristics. It overcomes the failure of traditional 2D/3D cell culture models and preclinical animal models to completely replicate the complex TME of human tumors. As a brand-new technology, OoC is of great significance for the realization of personalized treatment and the development of new drugs. This review discusses the recent advances of OoC in cancer biology studies. It focuses on the design principles of OoC devices and associated applications in cancer modeling. The challenges for the future development of this field are also summarized in this review. This review displays the broad applications of OoC technique and has reference value for oncology development.

## 1. Introduction

Cancer remains one of the leading causes of mortality worldwide [1,2]. According to the latest published data of the World Health Organization (WHO) and the IARC in GLOBOCAN, cancer accounts for one in six deaths globally, and the cancer burden will increase by 60% with estimated cases of 30 million by 2040 [3,4,5]. Although tremendous efforts have been made to improve cancer diagnosis and develop anti-cancer therapies over the past few decades, cancer remains a major issue worldwide. Due to weak efficacy or side-effect reactions, over 80% of drug candidates fail during the development stage, and few drugs are available to the market for clinical use [6,7,8,9]. Moreover, even if the drug has been approved for clinical application, it may be recalled for undisclosed adverse reactions, such as severe heart, kidney, or liver toxicity, which pose a severe threat to patients’ health [10]. The major reason for these issues is that human diseases and their treatment methods are mainly studied through in vitro tissue culture and animal models. However, the effect of drug toxicity on humans cannot be directly verified by these methods. Therefore, there is a lack of preclinical cancer models that can simulate human cancer’s complexity.

Two-dimensional (2D) platforms are the most commonly used models in vitro due to their relatively simple cell culture procedure, low cost, and availability to high-throughput drug screening and toxicity studies [11,12,13,14,15]. However, 2D cell culture models are relatively simple compared to tumors, failing to accurately reproduce the complicated three-dimensional (3D) tumor microenvironment (TME) [16,17,18]. Moreover, 2D platforms are not ideal for studying cell signal transduction mechanisms, chemical gradients, spatial structure changes, and drug resistance. Thus, results collected from 2D cell culture methods can be misleading for predictions of their application in vivo. Recently, 3D cell culture technology has gained more attention from researchers. Three-dimensional cell culture models, including tissue explants, spheroids, and transwell-based models, can accurately mimic the cell behavior, morphology, and physiology of 3D tumors, providing more realistic TME and predictive ability. Nevertheless, the 3D models cannot reproduce certain mechanical cues, such as hydrostatic pressure and fluid shear stress [19,20,21]. The animal model is a gold standard in cancer biology, allowing investigations in the living system, which can imitate the TME to assess tumor growth and drug response in vivo. However, animal models for clinical applications are hindered by their high cost, low-throughput drug optimization, long-term engraftment, and ethical controversy [22,23]. In addition, only a few types of human cancers can be applied to obtain a patient-derived xenograft (PDX) to construct a patient-derived animal model. The reason is that there are differences between animal and human genes due to their species specificity, resulting in inconsistent responses of animals and humans to drugs in drug tests [24]. Therefore, there is an urgent need to develop models that can accurately mimic key features, structures, and crucial interactions between various cells of human organs to cope with the shortcomings of conventional cell and animal models (Table 1). These models are more reliable and predictable in reproducing human TME for further understanding the complexity of cancer and the impact of anti-cancer therapies on humans to develop effective anti-cancer drugs.

Recently, advanced OoC platforms with more sophisticated approaches have been developed to capture the features of 3D architecture and physiological TME in human organs on a chip in vitro. Such platforms aim to resemble their native functionalities to improve the screening of anti-cancer drugs and elucidate the mechanism of cancer biology [25,26,27,28,29,30,31,32,33,34]. OoC technology employs a microfluidic chip as the core, combining biology, materials science, and engineering to simulate the microenvironment of native tissue and organs, containing living cells, biological fluids, mechanical stimulation, and other elements in vitro [35,36,37,38,39,40,41,42]. Generally, OoC devices are fabricated by “soft lithography”, which duplicates the patterns of a silicon template by pouring the liquid polymer into the template to create cell array platforms [34,35,43]. Typical OoC platforms for studying the tumor progression and tumor response to therapies include microfluidic chips, cultured cell or tissue slices, physiological stimulation levels required for tissue maturation, inlet ports for liquid and gas exchanges, and ports for delivery directional microvalves of liquids and small molecules (in the case of multi-organ chips) throughout the microfluidic device, as well as outlet ports for removal of waste from the system and sensors or optics for result readout [44,45,46]. The assembled structure of the OoC device makes it easy to image the microchannel chip to observe cell response and tumor formation. However, on the other hand, collecting cell samples from the chip is difficult. The chip usually needs to be taken apart, and such a process can disrupt the cell environment and lead to cell damage. However, compared with conventional models in vitro, these platforms have the ability to accurately control the mechanical environment, morphological structure, and chemical transfer rate at the cell-scale resolution, reconstructing the physiological dynamic properties of tissues, such as nutrient transport, shear stress, physiological flow, and drug effects in tissues in a controllable environment [47,48]. OoC provides a platform for microenvironment creation, cell culture, organ simulation, and in vitro evaluation of organ tissues. As a disruptive technology, OoC technology is of great significance for the realization of personalized treatment. OoC technology disintegrates human organs and tissues and changes the precise diagnosis of the “human body” into the precise diagnosis of the “organ”, providing more effective and targeted treatment. On the other hand, OoC technology is meaningful for developing new drugs. The cells of OoC platforms are directly derived from human beings, which effectively avoids the species difference between humans and animals and can accurately evaluate the toxicity of drugs on humans. At the same time, through the precise design and control of microfluidic chips, it can also simulate multiple types of organ-specific disease states in vitro, which can almost truly reflect the dynamic change rule of drugs in the body and its influence on organs, so as to conduct mechanism research on disease pathology, therapeutic intervention efficacy, and potential off-target effects. Thus, the failure rate of the clinical development stage is effectively reduced. In addition, OoC models are more cost effective than traditional animal testing. OoC technology has developed rapidly on the basis of the need for alternatives to animal testing and the need for early detection of drug toxicity.

Currently, OoC technology has successfully established a variety of healthy or diseased tissue and organ models, such as heart, kidney, liver, and lung, for anti-cancer drug screening and toxicity testing. It is meaningful for preclinical cancer drug screening, disease model establishment, and personalized treatment. Guo and his partners developed a mini tumor OoC based on microfluidics for the evaluation of cancer immunotherapy [49]. This technology may be used in preclinical models to predict tumor responses to cancer immunotherapy, assisting in treatment decisions and enhancing patient survival. Schuster et al. developed an automated, high-throughput microfluidic 3D organoid culture OoC system that enables real-time analysis of organoids. This integrated platform improves organoids models to screen and mirror the treatment process of real patients with the potential to facilitate treatment decisions for personalized therapy [50]. Zhao’s group designed a microfluidic tumor OoC model with hemispheric wells to study the tumor targeting and anti-cancer efficacy of multifunctional liposomes [51]. It is a convenient and powerful platform for the rapid and reliable evaluation of cancer drugs. The main challenges for cancer research at the moment are to effectively build in vitro TME and explore effective models for revealing the mechanism of the tumor, screening anti-cancer drugs, and developing therapeutic methods. The emerging OoC provides a brand-new technology platform for the construction of TME in vitro that can reproduce the key structural and functional characteristics of tumors in vivo, deeply comprehend the mechanism of tumor evolution, realize the accurate screening of anti-cancer drugs and the development of new tumor treatment strategies, and improve the survival rate of cancer patients. In this review, we present the latest advances in OoC technology in tumor biology research and modeling applications by combing microfluidics, microfabrication principles and materials, and tissue engineering technologies (Figure 1).

## 2. Manufacture Methodologies

### 2.1. Designing Basis and Materials of the OoC Model

In order to construct a native tumor model, the crucial characteristics of the real tumor should be duplicated. A tumor comprises various types of cells in a dynamic TME, wherein a host of biophysical and biochemical cues dictate the responses of cells [52]. The TME plays a pivotal role in tumor initiation, progression, and drug resistance, and is a key point in cancer research [53,54]. The TME is a complex ecosystem consisting of various cellular and noncellular components, including cells, soluble factors, signaling molecules, the extracellular matrix (ECM), and mechanical cues, which regulate the proliferation, function, and fate of tumor cells through two-way communication (Figure 2A) [55,56]. The soluble factors in the interstitial fluid, cell–ECM, or cell–cell adhesion, and mechanical cues (fluid strength, interstitial flow, shear stress, ECM stiffness, and ECM composition) trigger cell signaling in the TME. The interaction and functional association of tumor cells with surrounding tissues can create a new pathological “organ” that changes continuously as malignant tumors progress and respond to therapies. Furthermore, tumors, similar to normal tissues, require oxygen, nutrient supply, and removal of metabolic waste through blood vessels. Due to the rapid growth of the primary tumor, the associated blood vessels cannot provide sufficient oxygen and nutrients in time, resulting in hypoxia within the tumor [57]. In turn, hypoxia in tumors affects tumor cell genotype selection, metabolic regulation in anaerobic glycolysis, prosurvival gene expression, and epithelial–mesenchymal transformation (EMT) [58]. Therefore, hypoxia is often considered as a mediator in cancer progression and treatment of drug resistance. Compared with traditional in vitro tumor models, the OoC, which combines microfluidic technology with cell biology, could control environmental factors and accurately mimic human tumors, biological environment, and functional units in vitro. It can also simulate tumor cell migration and invasion, intravasation and extravasation, angiogenesis, and the progression of lesions from early to late stages, including EMT, tumor cell invasion, and metastasis [59,60]. The advantage of microfluidic platforms in simulating hypoxia within the TME is that they can generate physiologically relevant hypoxia or oxygen gradients by modulating the gas permeability of the platform. In addition, the microfluidic platforms could also reproduce various other TME characteristics, such as mechanical factors, chemical gradients, and biochemical interactions of different cells.

The core technology of OoC models is the microfluidics technique, which can precisely control and manipulate microscale fluids, especially submicron structural fluids [42,61,62,63,64]. With the development of micro-nano technology, microfluidics technology has evolved into a division of microelectromechanical system (MEMS) devices specifically for liquid processing. Microfabrication techniques, including microcontact printing, bioprinting technology, replica molding, and soft lithography, are usually applied to fabricate OoC microfluidic devices (Table 2). On the basis of precise microfluidic designs, OoC models allow for the manipulation of fluids at ultralow volumes (i.e., nanoliter and below), which can be used to simulate shear stress, physiological flow, drug exposure, and nutrient delivery [36,65,66,67,68]. With the development of new biomaterials and bioprinting processes, 3D bioprinting technology has become a promising methodology that fabricates OoC devices. It enables OoC devices to accurately recreate the TME of human patients in vivo [69,70,71,72]. According to the applications of the 3D bioprinting technique, there are different types of bioprinting methods with different characteristics, including inkjet bioprinting, laser direct bioprinting, stereolithography (SLA) bioprinting, and extrusion bioprinting (Figure 2B) (Table 3). Three-dimensional bioprinting technology can simultaneously print various biofunctional materials and cell types on cell-compatible substrates and accurately control the biomaterials of the carrying cells. Moreover, the heterogeneous microenvironment and complicated 3D microstructures of tumors can be reconstructed by 3D bioprinting technology with high spatial resolution and good reproducibility. It has the potential to build a miniaturized, multi-organ bionic pathophysiology model by 3D bioprinting technology, as well as provide a precise mechanism research platform for personalized cancer treatment research (Figure 2C).

Several materials have been employed to fabricate microfluidic devices for different applications, including inorganic silicon, PDMS and PMMA polymers, organic paper, and glass [40,73,74]. Silicon is the commonly used material in manufacturing microfluidic devices, with the advantages of resistance to organic solvents, easy metal deposition, and thermal conductivity [40]. However, the hardness and opacity of the material make it unsuitable for microfluidic applications. Glass has similar characteristics to silicon, with additional high optical transparency and high-pressure resistance, becoming an important material for the preparation of microfluidic devices [75]. Nevertheless, its hardness and high cost limit its application in microfluidic device fabrication, motivating researchers to find alternative low-cost materials. Recently, paper, as a cellulose-based material, has gained attention as a medium for microfluidic device fabrication because of its portability, its low expense, and its feature of being chemically modified or compositionally altered on the surface [76]. The multi-layer and stacked 3D structures can be easily made with paper. The capillaries of microfluidics based on paper pull solution through devices, which has been applied in medical, biochemical, and forensic studies. However, the mechanical strength of paper-based microfluidic devices in the wet state is weakened, and the transparency is limited by thickness. PDMS is the most widely used material for manufacturing OoC microfluidic devices due to its easy access, low cost, transparency, flexibility, biocompatibility, and gas permeability. The PDMS microfluidic structure can be fabricated by molds prepared by photolithography or other processes [77,78,79,80]. The PDMS model forms a sealed microfluidic system by attaching itself to the glass substrate via a plasma process. The microfluidics prepared by PDMS are gas permeable. It allows gas to freely flow without external air between chambers. In addition, the air permeability of PDMS microfluidics can be easily adjusted by modifying its composition, and transparency can assist in sensor imaging. However, PDMS is hydrophobic, resulting in forming air bubbles in the microfluidic channels and absorbing small molecules of hydrophobicity in solutions, such as drugs and biomolecules, ultimately leading to results bias. Moreover, since the mechanical properties of PDMS will be changed by aging, it is not suitable for long-time storage.

### 2.2. Cell Culture

OoC technology combined with 2D cell line models has been commonly used for genome-level studies of tumor growth and drug sensitivity [81,82,83,84,85]. However, in 2D cell culture models, only on the side that is in contact with the culture surface will the cell adsorption occur. Even worse, when many cells isolated from tissues are placed on the planar cell culture surface, they gradually become flattened, divide abnormally, and lose their differentiation phenotype, leading to lacking tissue structure and complexity. Moreover, OoC technology combined with 2D cell line models cannot mimic many characteristics of tumor disease, including hypoxia, altered cell contact with each other, and metabolic reprogramming. Therefore, OoC technology combined with 2D cell line models often fails to build effective tumor biology models. When 2D models are used for preclinical drug development, there is a heavy reliance on animal models for bioavailability and toxicology studies. Compared to OoC technology combined with 2D cell line models, the system constructed by OoC technology combined with 3D cell culture models is closer to the microenvironment of tumors in vivo with regard to cell morphology, differentiation, proliferation, transfer, and migration (Figure 3A) [15,86,87,88]. In 3D cell culture models, cell adsorption can occur throughout the cell surface, having the ability to establish physiological differentiation functions of cell–cell and cell–ECM interactions, simulating the specificity of natural tissues. Another important physiological property of 3D cell culture models is that the cultured cells possess the appropriate cell polarity. Combing microfluidic and 3D cell culture techniques makes it possible to mimic the complex 3D tissues and in vivo organ physiological of tumors. In addition, since cell migration can lead to infiltration and extravasation events during metastasis [89,90], invasion and extravasation could also be simulated by such combined techniques.

Tumor spheroid is a 3D culture model available to recapitulate the interactions between a cancer cell and the neighboring ECM [91,92,93]. Such a model can produce oxygen gradients and nutrients to form necrotic cores. Therefore, the spheroid model is feasible to simulate the central regions of vascularized tumors [94,95,96]. Although spheroids can reproduce some of the characteristics of tumors, such as chemotherapy/radiation resistance, heterogeneity of tumor cells, and the invasion/migration process, these models also have limitations. First, the spheroids belonging to avascular tumors, which lack the structure and complexity of the vascular tumor in vivo, being unable to fully mimic the cellular phenotype spectrum in the tumor milieu [97]. Second, spheroid cultures are performed under static conditions, lacking mechanical factors, such as shear stress, physiological flow, drug exposure, and nutrient delivery, which hinder the studies of drug sensitivity and toxicity for long-term culture. Another important limitation is the difficulty of forming spheroids for many tumor types, especially tumors with highly aggressive phenotypes. It leads to the failure of tumor detection in these cultures (e.g., MDA-MB-231 breast cancer cell lines). To resolve these problems, OoC provides more complex tissue-based culture models that replicate key characteristics missed in conventional monolayer or spheroid cultures by combing 3D culture model with microfluidics (Figure 3B) [98]. For instance, a model of breast cancer invasion has been successfully constructed by combing 3D microfluidics and spheroid culture. This 3D microfluidic tumor model uses surface-tension pumping to realize sequential loading of cells at various time points [99]. The correlation between tumor invasion and cell migration experiments in breast cancer biology is confirmed by employing this breast cancer OoC device. It shows the good consistency of such device in vivo xenograft tumor models. Moreover, designed and fabricated microfluidics combing with biomaterials can be used to create engineered 3D human-specific models. For example, the microfluidic OoC platforms could be biofabricated with biomaterials, such as hyaluronic acid-based hydrogel [100], collagen hydrogel [101], GelMA matrix [102], and hierarchical hydrogel [103] embedding the cancer cells for self-assembling and tissue formation. Such platforms have been achieved to construct gut, liver, glioblastoma, blood, and lymphatic vessel pairs in the biofabricated microfluidic channels for cancer investigation and drug discovery. Additionally, the cell culture, on the basis of microfluidic technology combined with other technologies, can be used to meet some special needs. For example, the platform based on the amalgamation of microfluidics and micro-optical components is of great importance for optical sensing, fluorescence analysis, and cell detection [104,105,106]. The microfluidic device integrated with acoustics could achieve non-contact, non-invasive, adjustable, and precise cell manipulation. Such models are meaningful for rapid and high throughput drug screening and helpful in biological applications and research [107,108].

## 3. OoC for Tumor Modeling

### 3.1. OoC for Tumor Vascularization Modeling

In the circulatory system, the microvascular system plays a critical role in blood flow, providing nutrients and removing waste products from the body. Nearly all the tissues in the human body, including the malignant ones, depend on the delivery of oxygen and nutrients through the blood vessels to survive. The sprouting of new blood vessels from an existing vascular system is called angiogenesis, and the blood vessels formed from progenitor cells are called vasculogenesis (Figure 4(Aa)) [109,110]. The combination of angiogenesis and vasculogenesis represents the fundamental processes of new blood vessel formation, which is crucial for physiological processes such as pregnancy, wound healing, tissue homeostasis, and fetal development [111,112,113]. However, during malignant progression, tumors will directly incorporate the existing blood vessels to achieve their own vascularization, obtain oxygen and nutrients needed for growth, and eliminate metabolic wastes [114]. It also provides a pathway for metastasis, which accounts for 90% of cancer deaths [115]. In the treatment of cancer, both chemotherapy and immunotherapy require the blood vessels to deliver drugs to tumors [116]. However, since the tumor is random and chaotic, it is hard for the drug to be delivered to the entire tumor, and it is difficult for the concentration of the drug to achieve an effective concentration [117]. In addition, the drug may be flushed away by the tumor blood vessels when it spreads to the surrounding tissue. It seriously reduces the effect of the drug. Therefore, for the tumor niche, tumor-associated vasculature is an important component, as well as a promising therapeutic target. Antiangiogenic drugs have been developed widely for cancer treatment, but clinical trials have yielded mixed results and were often with only modest improvements in survival. Thus, in order to explore more options for effective tumor treatment, it is critical to elucidate treatment failure factors. Replicating the heterogeneity and complexity of the TME for mimicking physiological barriers to drug or gene delivery will help translate in vitro results into in vivo studies.

To build more realistic tumor models in vitro, endothelial cells and fibroblasts were co-cultured/tri-cultured with tumor cells to create tumor spheroids of heterogeneity [118,119,120]. In such models, instead of developing perfusable tumor vasculature inside, distinct cell types were simply rearranged and stratified into different layers. Currently, the most effective strategy to achieve tumor vascularization in vitro is to induce the interaction of tumor spheroids with existing nearby blood vessels and reconstitute TME [121,122,123,124]. There are three categories of vessel-tumor models based on the size of the vascular tissue. They are vessel-tumor models [121,125,126], single-lumen vessel-tumor models [123,127], and endothelial-tumor models (Table 4) [128]. The different relative sizes between tumor and vascular tissue in these three models mimic the interaction between tumor and blood vessels at different stages of cancer. A vascularized tumor spheroid-on-a-chip model was developed by Ma’s group to verify synergistic vasoprotective and chemotherapeutic effects (Figure 4(Ab)) [129]. They constructed a perfusion–perfused vascularized tumor spheroid-on-a-chip model. The model consists of a vascular bed and tumor spheroid. The vascular bed is derived from the lung fibroblasts of normal human, and the tumor spheroid is derived from the umbilical vein endothelial cells and esophageal cancer cells of a cancer patient. This model was used to mimic the in vivo TME and prove that the prolyl hydroxylases inhibitor dimethylallyl glycine can inhibit the normal vascular degradation and enhance the effect of the anticancer drugs cisplatin and paclitaxel on human esophageal cancer spheroidal cells at the same time. The platform has potential in anticancer drug evaluation and clinical personalized medicine. Kim et al. constructed a vascularized lung cancer model to assess the facilitated transport of anticancer drugs and immune cells in an engineered TME [130]. By controlling the interstitial flow direction, the presence and location of human lung fibroblasts, as well as the location of pulmonary cancer spheroids, are able to control the speed and direction of tumor angiogenesis. They made use of the perfusable vascularized tumor platform not only to screen the effect and deliver the anticancer drugs on tumor spheroids with a special focus on the role of sprouting capillaries but also to investigate immune cell trafficking through sprouting capillaries. Therefore, this vascularized tumor platform has the potential application in immune cell delivery and tumor capillaries. It is worth noting that the number of vascularized tumors generated each time on the basis of the current in vitro vascularized tumor models is limited, and the improvement of the reproducibility of this method is still required. In the future, the development of a standard automated microfluidic chip platform for vascularized tumor spheroid array generation and multi-drug high-throughput parallel screening will be desired.

Moreover, OoC has been applied in pulmonary hypertension (PAH) research. PAH is a severe vascular disease with high morbidity and mortality, which could cause right ventricular hypertrophy and heart failure by narrowing the pulmonary arteries and pulmonary arterioles [131]. While various traditional models (such as animal and cellular models) have been used to study the pathophysiology of PAH, investigate sex disparity in PAH, and monitor the effectiveness of PAH medication therapy, such models can only partially recapitulate the PAH pathological features. They are not suitable for combinatorial study design to understand intricate cellular processes implicated in PAH pathogenesis. Recently, microfluidic OoC models have been available for the fabrication of PAH-on-a-device models to deeply understand the under-investigated PAH disease. Al-Hilal and Ahsan et al. designed and fabricated a microfluidic PAH-on-a-chip to investigate the gender differences in PAH and the therapeutic efficacy of both approved and developing anti-PAH medications [132]. They recreated the pathophysiology of PAH on the device by growing three types of pulmonary arterial cells (PACs)—endothelial, smooth muscle, and adventitial cells—in microfluidics. This miniature device based on OoC has the potential to be utilized in order to evaluate a variety of well-established and emerging hypotheses about the pathophysiology and pharmacological treatment of human PAHs. Subsequently, Edel and his partners developed a biomimetic pulmonary artery (PA)-on-a-chip to study the molecular and functional changes in endothelial and smooth muscle cells of human pulmonary vascular in response to triggers of the disease and their response to drugs [133]. The model provides a brand-new, optimistic, and simpler method for researchers to investigate pulmonary vascular remodeling and boost PAH medication development. Lately, Wojciak-Stothard and his colleagues have presented an organ-on-chip model of pulmonary arterial hypertension that identifies a BMPR2-SOX17-prostacyclin signaling axis [134]. This model provides a convenient method for researchers to study pulmonary vascular remodeling and advance PAH drug development because it captures important alterations in the pulmonary endothelial phenotype necessary for the induction of SMC remodeling, including a BMPR2-SOX17-prostacyclin.

### 3.2. OoC for Onco-Immuno Modeling

Cancer is known as an immunogenic disease that can stimulate complex immune responses by activating immune-inflammatory and immune-suppressive signaling pathways [135]. Moreover, the tumor TME from the cancer-related patient is formed by the dynamic interaction of metabolism and immunity between precancerous and cancerous tumor cells and stromal cells, playing a critical role in host evasion, metastasis, governing growth, and drug response [136]. The TME itself plays a key role in influencing the interactions between the tumor and the immune system, as well as responses to immunotherapy (Figure 4(Ba)) [53,54,137,138] During tumorigenesis, the TME is disrupted, and cell-to-cell and cell-to-matrix interactions are also altered. These changes activate new signaling pathways, neovascularization, and dysregulated cell death resistance mechanisms. As immunity changes, circulating immune cells such as tumor-infiltrating lymphocytes (TILs) and peripheral blood mononuclear cells (PBMCs) are affected [139]. Compared to traditional chemotherapeutics, cancer immunotherapies have become a clinically validated therapy for many cancers with improved efficacy and reduced systemic toxicity. They allow a subset of patients with metastatic tumor disease to achieve long-term remission by fighting malignant cells with the host immune system [139]. However, cancer immunotherapy still faces challenges, such as overcoming treatment resistance and changes in patient response [140]. For immunotherapy, the TME serves as a dynamic entity that is characterized by the transport of a complex series of stromal cells, tumor cells, and immune cells across the endothelial barrier to the tumor site [141,142,143]. Given the importance of the TME in regulating immune cell function, more complex tumor models are needed to reproduce these multifaceted dynamics to elucidate response to immunotherapy and resistance mechanisms [144,145]. OoC systems combining malignant and immune components have the ability to mimic the dynamic interactions between the immune system and cancer cells, facilitating the development of precise immuno-oncology and effective combination therapies [146,147].

OoC combined with microfluidic technology has many advantages for exploring tumor–immune cell interactions. For example, microfluidics can capture fundamental features of the interactions of multiple cell types while allowing tight control and real-time monitoring of the microenvironment. Neutrophils, the most abundant type of leukocytes in the blood, are the first immune responders to infection and inflammation and have been shown to have properties that promote tumors and limit tumors. Chandrasekaran and his colleagues developed a novel and microfluidics-integrated 3D tumor tumor-immune microenvironment (TIME)-on-a-chip device. This device based on OoC technology could be used to study the effect of neutrophils on ovarian tumor cells during the inception of their collective 3D invasion (Figure 4(Bb)) [148]. This TIME-on-a-chip integrates the tissue-engineered permeable microfluidic channels and functional 3D spheroids, making it possible to mimic tumor vascular tissue and recreate neutrophil exotosis and NETosis functions in vivo in a rapid, reproducible way. This versatile platform has strong adaptability and can be used to increase analytical throughput for performing complex in vitro biomimetic assays, such as drug screening or assessment of cytotoxic properties of biochemical molecules. Solid tumors create an inhibitory environment that places a huge burden on the immune system. The ability of immune cells, such as T cells and natural killer cells (NK), to destroy cancer cells can be severely impaired by waste accumulation, nutrient depletion, pH acidification, and hypoxia. However, the specific molecular mechanisms that drive immunosuppression and the ability of immune cells to adapt to the suppressive environment have not been fully revealed. Beebe et al. fabricated a microfluidic OoC platform in vitro to study NK cell responses to tumor-induced suppressive environments (Figure 4(Bc)) [149]. The platform not only allows tumor and immune cell evolution to be easily monitored but also has the potential to mimic key environmental features by combining microfluidics and organic models. The results suggest that the inhibitory environment generated by the tumor gradually erodes the cytotoxicity ability of NK cells, which results in a decrease in the monitoring ability of NK cells and tumor tolerance. Furthermore, NK cell exhaustion lasted for a long time after NK cells were removed from the microfluidic platform. Finally, NK cell exhaustion was alleviated by the addition of checkpoint inhibitors and immunomodulators.

### 3.3. OoC for Tumor Hypoxia Modeling

In disease, especially in cancerous tissue, changes in oxygen levels are prevalent due to the irregularities of blood vessels. This causes normoxia, hypoxia, or even almost complete hypoxia in some areas [150,151]. Hypoxia exists in 50–60% of solid tumors, accounting for 19–70% of tumor volume [152,153]. It has long been recognized as the main cause of drug resistance and promotion of metastasis since 1950 and has been associated with metastatic progression, poor prognosis, and tumor aggressiveness, resulting in shortened patient survival [154,155,156]. To be specific, gradients in oxygen tension are observed from vessels to distant regions [157]. Cells adjust their metabolism accordingly to adapt to such changes in oxygen availability, increase glycolytic behavior, and accumulate waste in large quantities in areas remote from blood vessels [158]. In addition to adapting to their metabolism, cells in these remote areas exhibit additional features of phenotypic changes: they do not proliferate like cells near blood vessels. Instead, they express different kinds of membrane proteins [159] while becoming stationary or even apoptotic and necrotic. In particular, such a phenotypic shift is accompanied by changes in cellular response to chemotherapy; radiotherapy; and, more recently, immunotherapy treatments [160,161,162]. Moreover, its invasiveness and metastatic potential are significantly increased. For instance, soft tissue sarcomas (STSs) tend to form immense and hyperxic tumors, leading to an increasing risk of metastasis [163,164,165]. Hence, to mimic the situation in vivo accurately and include such changes in cellular phenotype, differences in oxygen tension need to be controlled in the vitro tumor model design.

One of the key features of the TME is hypoxia. In tumors, hypoxia reduces the effectiveness of chemotherapy and radiation therapy [166]. The effects of radiation therapy (RT) are directly associated with hypoxia, as the permanent repair of DNA damage caused by radiation-therapy-induced free radicals requires oxygen [153,167]. It results in an oxygen enhancement ratio (OER) of 2–3:1, which means the radiobiological effect is reduced by 2–3 times in hypoxic cells compared to normoxic cells [168]. In addition, hypoxia can alter the metabolism of tumor cells and cause drug resistance [169]. The inability of abnormal blood vessels to properly deliver cancer therapeutics to the hypoxic core is the root cause of drug resistance. Finally, immune cells are unable to survive in a highly acidic hypoxic microenvironment, which promotes immune resistance. Currently, there are no effective and specific treatments for hypoxic cells in tumors. Such therapy development relies heavily on the availability of in vitro models, which can reproduce hypoxia in TME accurately.

The OoC platform is a miniature 3D human microfluidic tissue that is used as an organ-level model for the simulation of the key biological parameters and functions of relevant living models. Compared to traditional formats, microfluidic technology enables precise control at the micrometer scale over all chemical and physical parameters of cell culture because of the laminar flow nature, reduced device size, and the larger surface volume ratio of size-controlled 3D cell culture that has medium-to-high throughput (Figure 4(Cb)) [170]. Various models of cancer metastasis based on OoC platforms have been proposed for molecular mechanism study and drugs screening (Figure 4(Ca)) [171]. However, there are few models for hypoxia study. It is mainly because of the fact that the artificial induction of hypoxia brings about many challenges in cell culture and the maintenance of hypoxia [164,170,172,173]. Gervais’s group reported hypoxic jumbo spheroids-on-a-chip based on OoC technology to evaluate the treatment efficacy [163]. A microfluidic platform was manufactured by using soft lithography. It allows a maximum of 240 naturally hypoxic tumor spheroids cultured in an 80 mm × 82.5 mm chip. Through histopathology, the response of combined radiotherapy (RT) and the hypoxic prodrug tirapazine (TPZ) to giant spheroids generated by STS117 and SK-LMS-1 sarcoma cell lines were investigated. The results demonstrated that the microfluidic device and giant spheroids are powerful preclinical tools to study hypoxia and how it impacts the therapeutic response. Zhang et al. developed an oxygen-concentration-controllable 3D culture multiorgan microfluidic (3D-CMOM) platform for researching hypoxia-induced lung cancer liver metastasis and screening drugs [170]. The platform monitored the regulation ability of dissolved oxygen concentration using an oxygen sensing system and analyzed the performance of oxygen concentration regulation in a 3D cell culture chamber. In addition, the effect of lung cancer metastasis on the liver was investigated by the hypoxic microenvironment 3D-CMOM platform. The role of cancer cells co-cultured with fibroblasts in cancer metastasis was investigated by transcriptomics (RNA-seq) and protein expression detection. The oxygen-controlled 3D-CMOM provides a platform for exploring the hypoxia-induced tumor metastasis mechanism and hypoxia-related targeted anticancer drug effect.

### 3.4. OoC for Tumor Metastasis Modeling

Despite the significant progress achieved in saving the lives of cancer patients, metastasis remains the main course of cancer-related death. It accounts for approximately 90% of cancer deaths worldwide. Instead of the primary tumor, the secondary tumors that are formed by a complex metastases process [172] are the cause of most cancer deaths. This process begins in the primary tumor, which secretes exosomes and soluble factors into the bloodstream (Figure 4D) [173,174]. These signals propagate to the common site of metastasis, where they are thought to initiate tissue support for new tumor lesions. Then, primary tumor cells acquire the phenotypes of invasion and migration through epithelial–mesenchymal transition (EMT). The endothelial barrier of blood and lymphatic vessels can be penetrated by these cells and travel as circulating tumor cells (CTCs) in the body. CTCs that survive in the circulation can penetrate through the vascular endothelium back to a second organ—usually the liver, lung, brain, or bone (Figure 4(Da)) [175,176,177,178]. After infiltrating new organs, CTCs undergo an EMT that promotes the survival of CTCs in the tissue parenchyma [177,179]. Cells can also survive in a semi-dormant state until remodeling occurs [179]. Clues from the TME, such as cell populations, ECM and tissue (fluid) mechanics, and biochemical composition, have been shown to work as critical roles in metastasis development. Dissecting these cues from the TME in a controlled manner is challenging but important for understanding metastases and avoiding cancer progression by effective inhibition of the growth of primary cancer cells at metastatic sites.

Recently, OoC models have become a tool for the study of the TME and metastasis [180,181] (Figure 4(Db)). These models are based on microfluidic chips, containing cell culture chambers that are able to control the fluid flow, local gradients, the composition of the local environment, and tissue mechanics. For example, Wang’s group developed a metastasis-on-a-chip model incorporated with organ-specific ECM for simulating the kidney cancer progression in the liver in order to predict therapeutic efficacy and evaluate dosage responses to anticancer drugs in a physiologically relevant liver microenvironment [182]. Lamghari et al. presented a metastasis-on-a-chip model that reproduces neuro-breast cancer crosstalk in the context of bone metastasis to explore the sympathetic regulation of bone metastases in breast cancer on the basis of a humanized OoC model (Figure 4(Dc)) [180]. In this study, a new 3D-printing-based multi-chamber microfluidic chip transfer platform was designed to combine three different human cell types: (1) osteophilic breast cancer cell variants, (2) sympathetic neural cells, and (3) human peripheral blood osteoclasts implanted into the bone matrix to reproduce the effects of sympathetic activation on the dynamic crosstalk that occurs between breast cancer cells and osteocytes in a fully humanized model. This work introduces an innovative and versatile platform that is able to explore novel mechanisms of intracellular communication in the bone metastatic niche. Zhang and his colleagues reported a hepatocellular carcinoma–bone metastasis-on-a-chip model for the study of thymoquinone-loaded anticancer nanoparticles [183]. The bioreactor contains two chambers—one of them can house the encapsulated HepG2 cells and the other one can simulate bone niches that contain hydroxyapatite (HAp). A microporous polymer membrane acting as a physical vascular barrier was placed on top of the two chambers. There was also a common vascular chamber above this membrane with the medium circulating during culture. This study showed that the hepatocellular carcinoma–bone metastasis-on-a-chip platform is able to mimic some important features of the cancer metastasis process, thereby confirming the potential for studying metastasis-related biology and improving anti-metastatic drug screening.

**Figure 4 biosensors-12-01045-f004:**
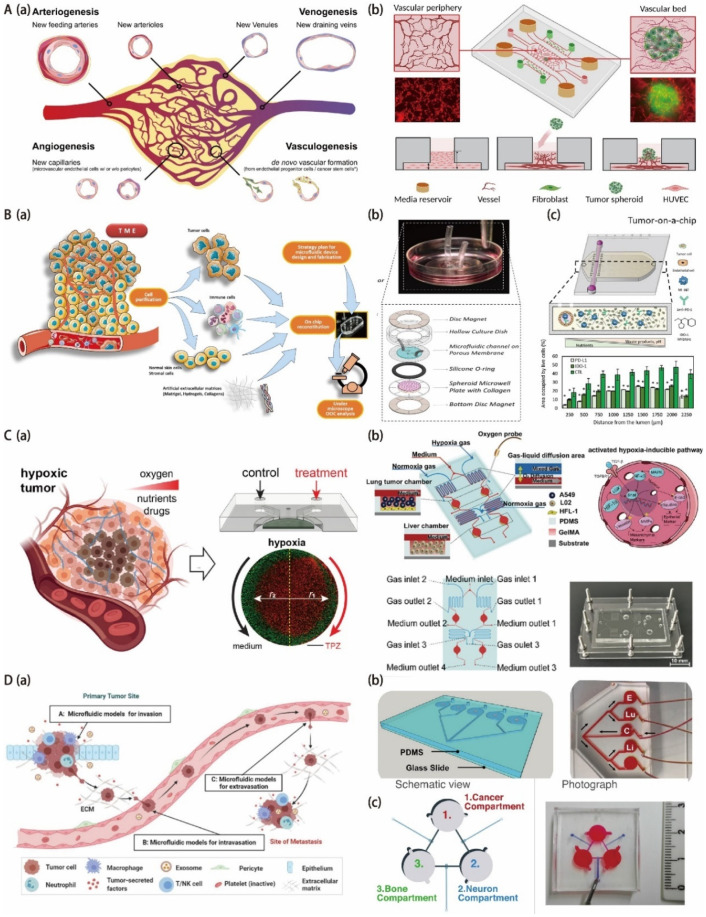
(**A**) Vascularized tumor chips. (**a**) The combination of angiogenesis and vasculogenesis represents the fundamental processes of new blood vessel formation. The newly formed blood vessels can provide nutrients and oxygen for malignant tumor development and metabolic waste removal. Moreover, they are able to have interactions with various types of cells in the vascular niche. The figure shows the processes of different blood vessel formations, including neoarteriogenesis, vascular remodeling, venogenesis, and angiogenesis. Adapted with permission from Ref. [110]. Copyright 2019, MDPI. (**b**) Schematic of the vascularized tumor spheroid-on-a-chip. The device consists of 5 microchannels in parallel by implanting HUVECs in fibrin gel into the chip (day 0) for self-assembling a vascular network (day 5), wherein tumor spheroids are placed and integrated with the surrounding blood vessels (day 15) to achieve tumor vascularization. Adapted with permission from Ref. [129]. Copyright 2022, ACS. (**B**) Onco-immuno chips. (**a**) Schematic representation of the reconstituted immuno-TME on OoC models. Adapted with permission from Ref. [138]. Copyright 2021, Frontiers. (**b**) Realization of TIME-on-a-chip on a 35 mm Petri dish platform: a ring magnet is used, microfluidic channels printed on a porous membrane are attached to a modified Petri dish and integrated with a microplate containing spheroids for mixing, surrounding the microplate O-rings, providing leak-proof components. Adapted with permission from Ref. [148]. Copyright 2021, IOPScience. (**c**) Schematic of the OoC for modeling the tumor microdevice. The bottom panel shows a cross-section of the microdevice. Endothelial cells (e.g., HUVECs) are lined in the lumen to generate vascular surrogates, allowing perfusion of culture medium, NK-92 cells, anti-PD-11 antibodies (e.g., atezolizumab), or IDO-1 inhibitors (e.g., epacadostat). Adapted with permission from Ref. [149]. Copyright 2021, AAAS. (**C**) Hypoxia chips. (**a**) Schematic of tumor hypoxia in vivo and recapitulating tumor hypoxia in vitro in microfluidic models with diffusion barriers. Reprinted with permission from Ref. [171]. Copyright 2022, ACS. (**b**) Schematic diagram and image of a 3D culture multiorgan microfluidic platform for precise control of dissolved oxygen concentration. Reprinted with permission from Ref. [170]. Copyright 2021, ACS (**D**) Tumor metastasis chip. (**a**) OoC models for mimicking tumor metastasis steps of (A,B) the invasion/intravasation process and (C) the extravasation process. Adapted with permission from Ref. [178]. Copyright 2022, MDPI. (**b**) A multi-site metastasis-on-a-chip microphysiological system for assessing cancer cells metastatic preference. Reprinted with permission from Ref. [181]. Copyright 2018, Wiley. (**c**) Schematic and photo of a metastasis-on-a-chip platform with three interconnected culture chambers to study the sympathetic regulation of bone metastasis in breast cancer. Reprinted with permission from Ref. [180]. Copyright 2022, Elsevier.

### 3.5. Cancer-Type-Specific Modeling by OoC

OoC is a state-of-the-art technology that can mimic organ and tissue function at the cellular level and achieve the replacement of healthy cells and associated ECMs in tissue-specific structures with cancer origins. Accurate modeling of the TME tissue-specific factors is essential to fabricate physiologically and clinically relevant in vitro platforms for tumor research (Table 5). Important aspects of the TME, including niche factors, biochemical gradients, complex tissue structures with tumor and stromal cells, and dynamic cell–cell and cell–matrix interactions can be ideally reproduced by OoC. Moreover, OoC can reproduce cellular confinement, which is a parameter of cell motility in the tissue stroma that is completely absent in the 2D analysis but critical for understanding the behavior of motile cells such as immune cells and cancer cells. OoC platforms based on microfluidic technology have successfully constructed various types of healthy and diseased tissues and organs, such as cystic fibrosis; microvascular obstruction; and heart, kidney, lung, pancreas, liver, skin, brain, eyes, gut, and neuropsychiatric diseases. This enables the recurrence of organ-level responses to drugs, toxins, cigarette smoke, radiation, pathogens, and normal microbes, as well as flow-circulating immunity. Therefore, the OoC platforms are becoming valuable tools for oncology research.

***Lung-on-a-Chip:*** The lungs are a key organ for exchanging gases between external oxygen and carbon dioxide in the blood. However, they are always at risk of being infected by aerosols due to inhalation of outside air [184]. As the most common cancer, lung cancer has the highest mortality rate of all cancers [3]. The metastasis of lung malignant cells to other organs is often observed in addition to solid tumors. The study of cell–blood flow, cell–gas flow, and cell–cell interactions in the respiratory tract has important implications for physiological studies and drug delivery (Figure 5(Ab)) [185,186]. Therefore, establishing an OoC model of lung cancer is important for understanding the treatment and pathogenesis of lung cancer (Figure 5(Aa)) [187,188,189,190]. A typical lung-on-a-chip platform consists of two microfluidic channels, which are separated by a porous extracellular matrix, with lung tumor cells integrated into pulmonary epithelial cells and pulmonary microvascular endothelial cells distributed on both sides [191,192,193]. This model is able to simulate various physiological functions of the lung. After electroplating, a gas–liquid level is formed after the removal of the cell culture fluid from the upper layer. The nutrient feed is delivered through the microvascular lumen. Choi et al. designed a multi-sensor lung-tumor-on-a-chip platform for toxicity assessment and physiological monitoring with integrated biosensors. This platform provides a promising way to evaluate the cytotoxicity of novel drug compounds for future micro-physiological systems and the development of personalized medicine [194]. Inger’s group developed human organ microfluidic chip models to create in vitro human orthotopic models of non-small cell lung cancer, recapitulating orthotopic lung cancer growth, therapeutic responses, and tumor dormancy in vitro [193]. These results showed the potential to better understand the mechanisms of cancer control and to inspire the discovery of novel drug targets and anti-cancer therapeutics.

***Breast-on-a-Chip:*** Breast cancer (BC) is a major cause of death worldwide and remains the most common malignancy in women, which is currently a major health problem worldwide [195]. It is necessary to develop new tools and techniques for better diagnosing and treating BC. Meanwhile, it is also important to obtain a deep understanding of the molecular and cellular participants involved in the progression of this disease (Figure 5(Ba)) [196]. The development of breast cancer, similar to other tumors, is a complex multistep process with a high degree of molecular and morphological heterogeneity. Understanding the progression and underlying heterogeneity of breast cancer is important for addressing the mechanisms associated with challenging tumor invasion, metastasis, and drug action. Jiang et al. reported a novel 3D breast-cancer-on-a-chip platform, composed of uniformly sized multicellular tumor spheroids (MCTS), ECM, and a microvessel wall, for the therapeutic evaluation of nanoparticle-based drug delivery systems [189]. This microfluidic platform is able to evaluate the behavior of dynamic transport and in situ cytotoxicity in one system, providing a more accurate and less expensive in vitro model for rapid drug screening in preclinical studies. Furthermore, breast cancer patients with pre-existing cardiac dysfunction may contribute to varying degrees of chemotherapy-induced cardiotoxicity incidence [197,198]. Shin’s group developed a heart breast cancer-on-a-chip platform for disease modeling and the monitoring of cardiotoxicity induced by cancer chemotherapy (Figure 5(Bb)) [199]. The proposed model is promising in helping to establish the prediction and early detection of chemotherapy-induced cardiotoxicity for individual patients in the future.

***Other types of OoC models*:** Furthermore, there are some other OoC models that have also been successfully constructed for oncology research. For example, Fan’s group developed a 3D brain-on-a-chip platform with PEGDA hydrogel for biological applications such as drug delivery (Figure 5C) [200]. The mepitastatin and irinotecan were injected into the cells. The results demonstrated that the platform can be used in drug screening and release experiments as a glioma chip model. Lu and Wang et al. developed a 3D biomimetic liver-on-a-chip model based on a microfluidics-based 3D dynamic cell culture system that integrates key components derived from the decellularized liver matrix (DLM) with gelatin methacryloyl (GelMA) for toxicity testing (Figure 5D) [201]. This DLM-GelMA-based biomimetic liver-on-a-chip model can better simulate the TME in vivo. It has exceptional prospects for extensive pathological and pharmacological research. Chen’s group designed a biomimetic pancreatic cancer-on-a-chip model to reveal endothelial ablation via ALK7 signaling. This OoC model offers a valuable in vitro platform for understanding the PDAC-driven endothelial ablation process and proposes a possible mechanism for tumor hypovascularity. Habibovic and his colleagues developed a colorectal OoC system as a 3D tool for precision onco-nanomedicine (Figure 5E) [202]. This model is capable of reconstructing the physiological function of microvascular tissue, which is a tool to evaluate the efficiency of the dynamic controllable gradient delivery of antitumor drug nanoparticles through the core chamber of a microfluidic chip. This gradient is provided by perfused side channels that mimic microvessels. The results show that the 3D platform works better for efficacy/toxicity screening in a more physiological setting. To replicate the in vivo microenvironment of lung cancer metastasis, Wang’s group designed and constructed a multi-organ microfluidic chip [203]. This multi-organs-on-a-chip model was composed of an upstream “lung” and three downstream “distant organs” that are able to reproduce the tissue–tissue interfaces and complex functions of lung and distant organs. By using this model, the metastasis of lung cancer to the bone, liver, and brain was explored. Cell–cell interactions and cell physiology can also be analyzed in a more physiologically relevant context. This system offers a valuable tool to simulate the vivo microenvironment of cancer metastasis and explore cell–cell interactions during metastasis. In general, the OoC model based on microfluidics can be extremely beneficial in cancer research.

**Figure 5 biosensors-12-01045-f005:**
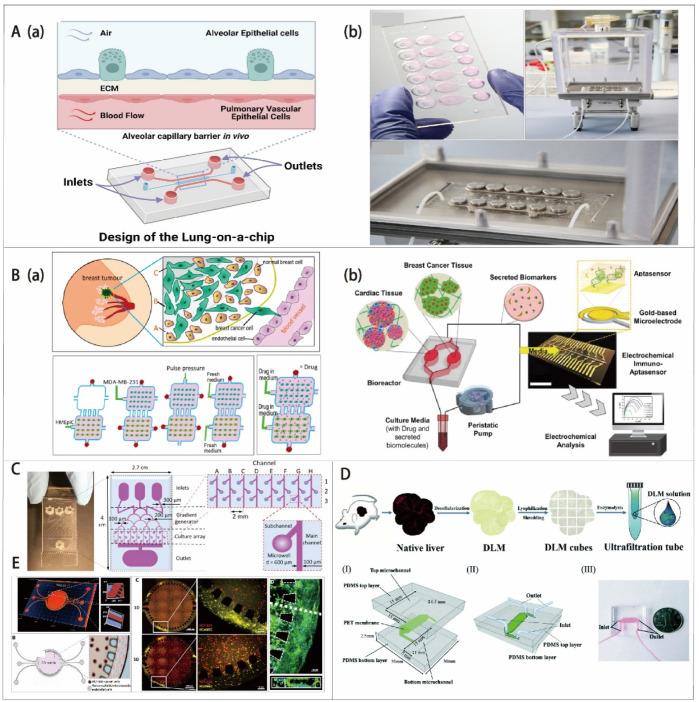
(**A**) Design and model of lung-on-a-chip. (**a**) Cross-sectional view of a lung-on-a-chip microfluidic model with two distinct channels separated by a thin porous membrane. Reprinted with permission from Ref. [190]. Copyright 2022, Elsevier. (**b**) The model of lung-on-a-chip comprising six wells is used to mimic the lung alveolar barrier, whereby cells are cultured directly at an air–liquid interface for inhalation assays. Reprinted with permission from Ref. [185]. Copyright 2019, Elsevier. (**B**) Breast OoC. (**a**) Schematic illustration of the metastatic breast tumors and on-chip steps for cell loading and co-cultivation as well as drug treatment. Adapted with permission from Ref. [196]. Copyright 2016, Nature-Springer. (**b**) The patients of breast cancer with preexisting cardiac dysfunctions may lead to different incident levels of chemotherapy-induced cardiotoxicity (CIC). This heart breast-cancer-on-a-chip platform with iPSC-derived cardiac tissues and BC tissues could be used for disease modeling and monitoring of cardiotoxicity induced by cancer. Reprinted with permission from Ref. [199]. Copyright 2020, Wiley. (**C**) Schematic of brain cancer chip for high-throughput drug screening. This chip has a gradient generator with a Christmas-tree-shaped channel system. The channel width is gradually reduced from 300 μm to 100 μm. Moreover, it also has an array of 24 independent culture chambers with 3 inlet banks and 1 outlet bank. Subchannels connect the microwells to the main channel and prevent captured cells from escaping the microwells. Adapted with permission from Ref. [200]. Copyright 2016, Nature-Springer. (**D**) Schematic of the decellularized liver matrix-based liver OoC, including use of a natural liver to prepare the DLM solution and a 3D schematic diagram of the equipment components, consisting of the microchannels from the top and bottom, the PET membrane, and the air inlet and outlet. Adapted with permission from Ref. [201]. Copyright 2018, Royal Society of Chemistry. (**E**) Design and characterization of the colorectal OoC system microfluidic chip for precision onco-nanomedicine. Adapted with permission from Ref. [202]. Copyright 2019, AAAS.

**Table 5 biosensors-12-01045-t005:** Summary of the different types of OoC for tumor models.

OoC for Tumor Models	References	Cell Types	Drugs	Applications
Tumor vascularization model	[129]	Human esophageal carcinoma (Eca-109)	Paclitaxel Cisplatin	Simulate the TME in vivo and demonstrate that the PHD inhibitor dimethylallyl glycine prevents the degradation of normal blood vessels while enhancing the efficacy of the anticancer drugs paclitaxel and cisplatin in Eca-109 spheroids.
	[130]	HUVECs Lung cancer cells (A549)	Doxorubicin-HCl (DOX)	Build a vascularized lung cancer model to evaluate the promoted transport of anticancer drugs and immune cells in an engineered tumor microenvironment.
	[132]	PAH-ECs PAH-SMCs PAH-ADCs		Elucidate the sex disparity in PAH. Study the therapeutic efficacy of existing and investigational anti-PAH drugs.
	[133]	HPAECs HPASMCs		Study pulmonary vascular remodeling and advance drug development in PAH.
	[134]	HPAECs HPASMCs		Elucidate the sex disparity in PAH. Study the therapeutic efficacy of existing and investigational anti-PAH drugs.
Onco-immuno model	[148]	OVCAR-3 cells		Construct tumor-immune microenvironment (TIME)-on-a-chip to mimic 3D neutrophil–tumor dynamics and neutrophil extracellular trap (NET)-mediated collective tumor invasion.
	[149]	Breast cancer cells (MCF7)		Study how NK cells respond to the tumor-induced suppressive environment.
Tumor hypoxia model	[163]	SK-LMS-1, and STS117 cells	Tirapazamine (TPZ)	Provide an OoC platform for allowing easy culture, maintenance, treatment, and analysis of naturally hypoxic sarcoma spheroids.
	[170]	A549 HFL-1 Human normal liver cells (L02) cell lines		Providing an oxygen-concentration-controllable multiorgan microfluidic platform for studying hypoxia-induced lung cancer-liver metastasis and screening drugs.
Tumor metastasis model	[180]	Human CD14+ monocytes MDA- 1833 henceforth SH-SY5Y (ATCC) cells		Explore the sympathetic modulation of breast cancer bone metastasis.
	[182]	HepLL and Caki-I cells	5-FU-loaded PLGA-PEG NPs	Provide a novel 3D metastasis-on-a-chip model mimicking the progression of kidney cancer cells metastasized to the liver for predicting treatment efficacy.
	[183]	Human HepG2 HCC cells	Thymoquinone-loaded anticancer nanoparticles	Model and track hepatocellular carcinoma (HCC)–bone metastasis. Analyze the inhibitory effect of thymoquinone in hindering the migration of liver cancer cells into the bone compartment.
Lung-on-a-chip	[193]	NSCLC cells	Tyrosine kinase inhibitor (TKI)	Develop an OoC device to recapitulate orthotopic lung cancer growth, therapeutic responses, and tumor dormancy in vitro.
	[194]	NCI-H1437 lung cancer cells	DOX	Develop a multi-sensor lung-cancer-on-a-chip platform for transepithelial electrical (TEER)-impedance-based cyto-toxicity evaluation of drug can-didates.
Breast-on-a-chip	[189]	HUVECs Human breast cancer cell lines T47D and BT549	CDs-PEG-FA/DOX	Construct a 3D breast-cancer-on-a-chip for the evaluation of nanoparticle-based drug delivery systems.
	[199]	Breast cancer spheroids (SK-BR-3) iPSCs		The real-time drug delivery monitoring and in situ cytotoxicity assays in one system. Provide a heart-breast OoC platform for disease modeling and monitoring of cardiotoxicity induced by cancer chemotherapy.
Brain-on-a-chip	[200]	Glioblastoma cells (U87)	Pitavastatin Irnotecan	High-throughput drug screening. Mimic TME.
Liver- on-a-chip	[201]	HepG2 cells	Acetaminophen	Toxicity testing.
Pancreatic-cancer-on-a-chip	[202]	HCT-116 cells (human colon cancer cell line)	CMCht/PAMAM dendrimer Nanoparticles	Assessment of precision nanomedicine delivery.
Multi-organ microfluidic chip	[203]	16HBE Human non-small cell A549 HUVECs WI38 THP-1 HA-1800 Fob1.19 L-02		Mimic lung cancer metastasis to the brain, bone, and liver.

## 4. Conclusions and Future Perspectives

Cancer has been a major cause of death worldwide for decades. Although the mechanisms of cancer formation, metastasis, and treatment have been studied for years, some questions still remain unresolved. The OoC system has the potential to bridge the gap between traditional in vitro cell culture and in vivo experiments and to accelerate in vitro cancer research. Compared to traditional techniques, microfluidic platforms offer many advantages, including high sensitivity, low cost, adjustable flow, short processing times, high-throughput screening, and reduced requirement of samples and reagents. To promote the development of new OoC models, microfluidics technology was introduced with the aim of reliably reconstructing an in-vivo-like microenvironment for cancer research. The OoC platforms based on microfluidic technology can simulate the main TME characteristics in vivo, including cell–cell and cell–matrix interactions, physical and chemical gradients, hypoxia, and vascularization. As an alternative preclinical model to the traditional ones, it shows a more realistic and accurate evaluation prospect in the study of metastasis, tumor distribution, and growth mechanisms, as well as drug toxicity and therapeutic effects. OoC models can capture key features of real tumors and be used for diagnosis and treatment while minimizing the need. At present, plenty of OoC platforms have been designed and manufactured, such as liver, lung, brain, and breast tumor models. They are mainly applied to the screening of anti-cancer drugs and the basic research of cancer metastasis. One of the OoC’s ultimate goals is to establish an artificial tumor model by simulating physiologically relevant environments with controlling material, dimensional, and microenvironment variables (including chemical and mechanical factors) for animal and human trials.

While OoC technology has made good progress in reconstructing tumor microenvironments in vitro, it still faces many challenges before it can be widely integrated into the actual pharmaceutical industry and clinical applications. First, for the simulation of in vivo conditions of organisms, the biocompatibility and hardness of materials will be a problem that affects cell culture. The flexibility and high air permeability of the PDMS make it the most used material for the preparation of organ chips. However, PDMS can adsorb hydrophobic compounds, including proteins and drugs, easily, as well as being able to reduce the effective concentration and activity of drugs. Such absorption can lead to experimental errors, limiting its application. Therefore, it is necessary to explore new biocompatible materials for OoC manufacture. In addition, prior to the manufacture of this device, it is needed to make further consideration of the specific TME pathophysiology. Such pathophysiology requires the integration of various biophysical cues in a physiologically relevant manner, including elasticity, matrix stiffness, and many other biochemical factors. Second, although the OoC platforms can simulate the in vivo TME and even build multi-organ tumor chips to reconstruct the interaction of adjacent tissues, simulating the various complex signaling functions of other non-neighboring organs of the human body to respond to cancer is challenging. Third, although imaging on a microfluidic chip in an OoC system makes it easy to collect cellular response and observe tumor formation, it is challenging to collect cell samples from the chip. It requires disassembling multiple chips during sample collection, which can easily contaminate the environment of the cell culture. At the same time, the sample may be damaged during the collection process. Experimental procedures such as immunohistochemistry will be hampered by such defects. Therefore, a more reliable microfluidic chip sample collection system is imminent. Lastly, OoC systems are based on esoteric micromachining techniques that require skilled and experienced researchers to fabricate microfluidic devices. However, limited researchers are proficient in micromanufacturing equipment and tissue engineering techniques. Therefore, developing user-friendly on-chip systems and standardizing data collected from different labs are meaningful for researchers across different fields to easily access these models. In summary, although the OoC platforms still face many challenges, they are still promising platforms for future cancer diagnosis and treatment. To accomplish these goals, interdisciplinary collaboration is required among researchers in areas such as biomedical engineering, materials science, biophysics, cell biology, and oncology. By designing and optimizing oncology-on-a-chip systems for the development of cancer research and drug invention, the bio-inspired design could be translated into clinical applications and impact diverse fields such as cancer research, micromachining, and machinery.

## Figures and Tables

**Figure 1 biosensors-12-01045-f001:**
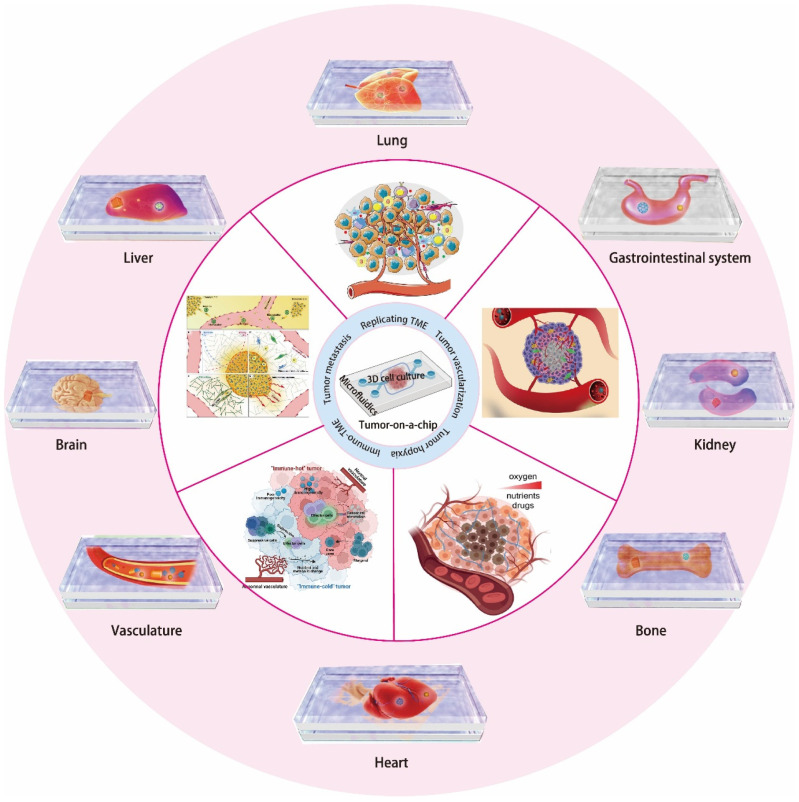
Schematic of OoC platforms for mimicking the TME and functions in vitro and their applications of reconstructing the organs on microfluidics for oncology studying.

**Figure 2 biosensors-12-01045-f002:**
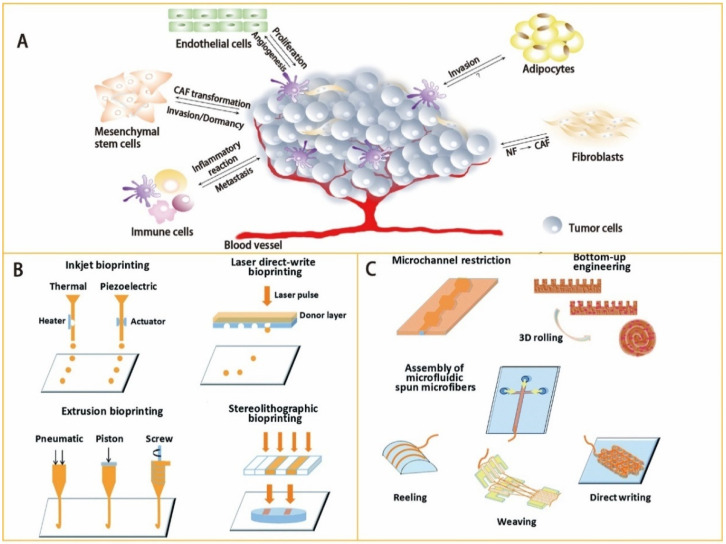
(**A**) The TME is a complex ecosystem consisting of various cellular and noncellular components, such as cancer cells, fibroblasts, multiple chemical factors, the extracellular matrix, the vasculature system, and mechanical cues. The progression of a tumor is critically influenced by the interaction between tumor and TME. These factors should be included in the construction of OoC models. Adapted with permission from Ref. [56]. Copyright 2019, MDPI. (**B**) Bioprinting techniques mostly used for the generation of microfluidics. Adapted with permission from Ref. [71]. Copyright 2018, Royal Society of Chemistry. (**C**) Microfluidic approaches involved in creating tissues/organs. Adapted with permission from Ref. [71]. Copyright 2018, Royal Society of Chemistry.

**Figure 3 biosensors-12-01045-f003:**
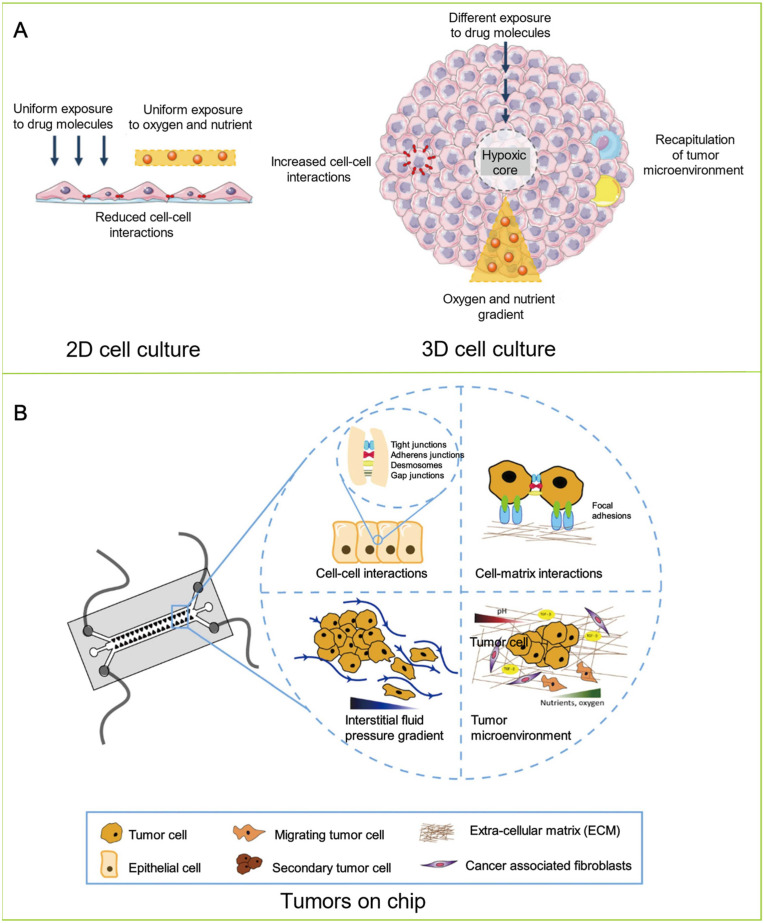
(**A**) The main differences between 2D and 3D cell cultures. Adapted with permission from Ref. [96]. Copyright 2020, MDPI. The monolayer formed by 2D cell culture can uniformly lead to exposure to oxygen, nutrient, and drug molecules. However, 2D cell culture models are unable to properly simulate the architecture and microenvironment of in vivo tumors. Compared to in vivo cells, the cells cultured by 2D cell culture models show fundamental differences in cell proliferation, morphology, differentiation, signal transduction, and metabolism. Three-dimensionally cultured cells are more similar to the in vivo environment in terms of cell proliferation, migration, differentiation, morphology, and transfer. In particular, spheroids are feasible to simulate the central regions of vascularized tumors since the model can produce oxygen gradients and nutrients to form necrotic cores. However, both 2D and 3D cell culture models are performed under static conditions, lacking mechanical factors, such as shear stress, physiological flow, drug exposure, and nutrient delivery. As a result, they hinder the studies of drug sensitivity and toxicity for long-term culture. Copyright 2020, MDPI. (**B**) The microfluidic platform is an effective tool to investigate a variety of key biological phenomena, from cell–ECM and cell–cell interactions to the flow of stroma within TME. Reprinted with permission from Ref. [98]. Copyright 2022, Elsevier.

**Table 1 biosensors-12-01045-t001:** Conventional pre-clinicals in cancer research.

Model	Advantages	Disadvantages
Two-dimensional cell culture (cell lines)	Simple and economic High-throughput drug screening and toxicity studies Reproducible and time saving	Oversimplified for tumors Low success rate for establishing tumor models Lack of tumor heterogeneity and TME
Three-dimensional cell culture	Recapitulate the architecture of tumors Retains tumor heterogeneity Progression of 2D cell cultures and in vivo models	High cost and time consuming Fails to represent the consequences of mechanical cues Unified methods of organoid production Tools to analyze them are limited
Animal	Gold standard in cancer biology Partly recapitulate the TME to assess tumor growth and drug response in vivo Allow investigation of a tumor in a living system	High cost, low-throughput, time-consuming engraftment and ethical controversy Species-specific differences leading to false drug test results Not all human cancers can be successfully transplanted to generate a patient-derived mouse model

**Table 2 biosensors-12-01045-t002:** The microfabrication methods for microfluidics.

Methods	Process Technologies	Advantages	Disadvantages
Photolithography	Lithography Etching	Precisely control the shape and size of the form High resolution	Time-consuming and expensive Requires many steps to generate one microfluidics device
Soft lithography	Self-assembled monolayers Elastomeric stamp Molding of organic polymers	Easy to replicate Allows the generation of multiple microfluidics devices of the same mold in a short period of time Reusable molds	Pattern deformation and vulnerability to defects Inappropriate for mass production
Replica molding	Using PDMS to make a negative embossed image of the master Cast prepolymer against PDMS master and generating the designed device	Easy to operate Mass productions No expensive equipment is needed	Casting material is limited High cost for mold fabrication Master needs to be prepared by photolithography and some other technologies
Microcontact printing	Stamp made by replica molding Self-assembled monolayer technology	Fast speed and low cost Simplicity of operation No need for a clean room Suitable for many different surfaces Flexible and changeable operation method	Mold deformation Substrate contamination Shrinkage and expansion of a stamp mold Fluidity of ink
Bioprinting technology	Extrusion-based bioprinting Inkjet bioprinting Stereolithography-based bioprinting Laser-assisted bioprinting	Rapid production and easy prototyping capability Control of complex 3D tissue geometry Precise and reproducible substrate and cell scaffold	Printing process can cause cellular damage Limited selection of material Unable to produce small features

**Table 3 biosensors-12-01045-t003:** Three-dimensional bioprinting technologies for organ-on-a-chip fabrication.

Three-Dimensional Bioprinting Technologies	Advantages	Disadvantages
Inkjet	High resolution (50 μm) High precision Fast printing speed Multiple reservoirs Prints multiple bioink simultaneously	Needle clogging at high-viscosity ink or high concentrations of cells Nozzle is easy to lose Make mechanical or thermal damage to cells Moderate cost for high-resolution systems
Extrusion	The viscosity of material from low to high can be printed Low cost Ease of use High mechanical strength	Moderate resolution (≈100 μm) High-throughput screening is limited by the speed of printing Shear forces may affect cell survival The selection of bio-ink needs to take into account of gelation, curing, shear thinning, and other properties
Laser direct	Non-contact manufacturing method No mechanical damage to cells High resolution (1–50 μm) Viscous or solid solution	Relatively high cost Limited materials for printing Limited degree of automation Difficult to print complex structure
LA	High resolution (3–300 μm) Fast speed Easy control of matrix Properties High cell viability	Poor hollow-structure capabilities Requires photo-curable bioink

**Table 4 biosensors-12-01045-t004:** Vessel-tumor models based on the size of the vascular tissue.

Models	Characteristics	Methodologies
Capillary vessel-tumor model	The diameter of formed vessels is about 10 μm	Co-culturing or tri-culturing the tumor cells with fibroblasts and endothelial cells (ECs). Vessel network formed by endothelial cells self-assembling.
Single-lumen vessel-tumor model	The diameter of the single-lumen vessel tubes is around several hundred micrometers	Prior to tumor spheroid/organoid seeding, ECs were covered in a pre-formed hollow channel to form a vascular channel.
Endothelial-tumor model	Monolayer endothelium-tumor co-culture model Microfluidic version of “trans-well assays”	The model is composed of upper and lower two-tier structures. The upper and lower layers are usually separated by a porous membrane. The endothelium and epithelium are seeded on the upper and lower sides of the membrane.

## Data Availability

Datasets generated during and/or analyzed during the current study are available from the corresponding author on reasonable request.

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
