# Peer review of "Recent Advances of Organ-on-a-Chip in Cancer Modeling Research"

_biosensors, 2022, doi:10.3390/bios12111045_

Round 1

Reviewer 1 Report

This manuscript summarizes the application of OoC technology in cancer models and the advantages of using microfluidic technology in OoC, and gives a detailed overview from the research background, research progress and application, analyzes the existing technical problems, and looks forward to the future development direction. In order to make the article more complete, we put forward the following suggestions.

1.       The manuscript introduces the application of organ chip in the establishment of cancer in vitro model in detail, but the introduction lacks the social significance of studying this technology. If the author can highlight its advantages and research significance, it will improve the integrity of the article.

2.       In the third part of the manuscript, the author introduces different types of tumor models based on OoC. If a table is established to classify and compare the contents of this part, the article will be more organized.

3.       In the manuscript, the author introduced the application of microfluidic technology in organ chips in great detail, but lacked a standardized, clear and reasonable summary of overall knowledge (technical methods). If the author's summary can be added, it can be further improved scientifically.

4.       If the author can briefly explain the progress and application of organ-on-a-chip in the treatment of other diseases such as pulmonary hypertension, the article will be more complete.

5.       The last paragraph of the manuscript explains three shortcomings of OoC, the third of which is that it is difficult to collect samples from the chip. This problem was not pointed out when comparing OoC with traditional in vitro cell culture and in vivo experiments in the front of the manuscript. If you can briefly explain this in the front of the manuscript, it will help to improve the structure of the article.

6.       We suggest adding some Organoid research based on microfluidic technology, such as acoustic wave based, to the manuscript to make the article more comprehensive. See Lab Chip, 20, 3515, 2020. Lab Chip, 21, 4005, 2021

Reviewer 2 Report

The manuscript entitled " Recent advances of organ-on-a-chip in cancer modeling research" by Liu and colleagues is timely and well-written. The authors discuss the different technologies for organ-on-a-chip fabrication, cell culturing options and methodology, and examples of tumor-on-a-chip models.  

Below are some comments and questions that need to be addressed to improve the article.

1) Page 6, line 169. There is a typo "(low/high?)"

2) Figure 3, line 248. Typo "f". Line 258, the sentence for Copyright is in bold, but not in line 260

3) Cell culture section 2.2- The authors focused only on tumor spheroids for culturing cancer cells in 3D. They explained that spheroids have limitations, and one of the most important limitations is "the difficulty of forming spheroid for many tumor types, especially tumors with highly aggressive phenotype" (pages 233-234). This limitation can be solved using biomaterials. They do not mention the possibility of designing and fabricating a biomaterial to embed the cancer cells for self-assembling and tissue formation. Using biomaterials coupled to a chip is an alternative to be mentioned in this review. Both strategies obtain relevant results in terms of biomimicry and drug responses. See for example PMID: 33041741, PMID: 26888480, PMID: 28062831 or PMID: 35028584.

4) Page 10, lines 288-290. Could the authors justify why they sustain that "inducing the interaction of tumor spheroids with existing nearby blood vessels by reconstituting TME is the most effective strategy to achieve tumor vascularization in vitro". I suggest comparing the other strategies for vascularization and explaining the pros and cons of each.

5) page 10, lines 290-294- "There are three categories of vessel-tumor models based on the size 290 of the vascular tissue. They are vessel-tumor models[107,111,112], single-lumen vessel- 291 tumor models[109,113], and endothelial-tumor models[114]". 

I suggest explaining these models better, preparing a table summarizing the bibliography, and indicating the methodology behind the different model systems. 
